# Speaker-Invariant Speech Recognition through Fine-Tuning on Individual-Specific Data with Voice Conversion

## Abstract

In this paper, we propose a speaker-invariant speech recognition method that fine-tunes a pre-trained model (Obtained by a self-supervised learning method) on a selected subset of data containing speech from a specific individual. This fine-tuning changes the network's behavior, allowing it to focus on information that is important for tasks such as ASR and phoneme recognition while reducing sensitivity to speaker-specific vocal characteristics. In the test time, we recommend employing voice conversion techniques to transform the voices of diverse individuals to match that of the individual used for training.

## 1 Introduction

Recent advancements in deep learning and the abundance of data have significantly improved the performance of automatic speech recognition (ASR) systems. However, labeling data for tasks such as ASR remains a challenging endeavor, particularly when compared to tasks such as image classification. With an enormous amount of unlabeled audio sources available on the internet, it is essential to find efficient ways of utilizing this data. Insights from the language learning process in children can provide a valuable perspective on this issue. Children acquire semantic information and language structure through interactions with their environment, followed by supervised spelling education from teachers. Inspired by this approach, the objective is to learn speech representation through self-supervised methods. These methods must be carefully designed to extract robust representations that can effectively support various downstream tasks, including speaker recognition, phoneme recognition, and more. In this paper, we propose a novel approach that addresses the issue of sensitivity to speaker variations in pre-trained speech representation learned by the self-supervised learning method for ASR downstream task.

### 1.1 Related Works

Self-supervised methods for speech representation learning typically involve two stages. During the first stage, the model is pre-trained via the design of a pretext task, which enables the acquisition of high-quality speech representations. In the second stage, these learned representations are utilized for downstream tasks, where the model is trained in a supervised manner using task-specific labels. Pretext tasks can be categorized into three types: generative, contrastive, and predictive. Generative methods involve reconstructing one part of the data using another part, and recent works employing this approach include Yue & Li (2021), Ling & Liu (2020), Liu et al. (2021), Liu et al. (2020) and Chung et al. (2019). Contrastive methods, on the other hand, involve bringing the anchor closer to positive samples while moving away from negative samples, and some notable works that have utilized this method include Chung et al. (2021) and Baevski et al. (2020). Predictive methods have a distinct model for target learning and follow the teacher-student approach. In the predictive method, the most famous works are Hsu et al. (2021) and Chen et al. (2022). The choice of pretext task is critical as it can significantly impact the learned representations' quality and downstream task performance.

## 2 METHODOLOGY

In the conducted research, the information encoded in the speech representations obtained by the self-supervised methods has a behaviour similar to the encoder-decoder. Speaker-related information is encoded in the primary layers. This information is suitable for tasks such as speaker recognition and diarization. As we get closer to the middle layers, first phonetic information and then words are captured, which are important for ASR and phoneme recognition tasks. Of course, this behaviour is not unpredictable because, for example, the generative method deals with data reconstruction. In the end, the last layers return to the original state and become like audio data. Baevski *et al.* Baevski et al. (2021) have demonstrated the best results were produced when the phoneme category was seen on the middle layers of learning. Pasad *et al.* Pasad et al. (2021) show that fine-tuning changes the model's behavior and focuses on information that is important to it. Since a task like ASR requires information about phonemes and words, they should be robust to different speakers, or in other words, speaker-invariant. An analogous concept in computer vision is shift-invariant processing, which allows convolutional neural networks to effectively recognize objects even when they are translated or rotated within the image. To achieve this goal, we propose fine-tuning a pre-trained network on a selected subset of data containing speech from a specific individual. By fine-tuning the network in this manner, we can reduce its sensitivity to the speaker's unique vocal characteristics, and instead leverage its capacity to accurately recognize words and other linguistic features. Since the network fits on the voice of this particular person, we recommend employing voice conversion techniques to transform the voices of diverse individuals to match that of the individual used for training.

## 3 CONCLUSION

In this paper, we first investigated the behavior of speech representations extracted by self-supervised learning methods from unlabeled raw audio, and then based on this behavior, we proposed a method that reduces the speaker information for the pre-trained network and focuses on higher-level information in ASR task.

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
