# OpenReview forum: "Speaker-Invariant Speech Recognition through Fine-Tuning on Individual-Specific Data with Voice Conversion"
_ICLR.cc/2023/TinyPapers — Submitted to Tiny Papers @ ICLR 2023_

### Official Review · Reviewer_xKVm · 2023-03-24

**Confidence:** 4

**Summary Of Contributions:**

The authors present a research proposal for creating a speaker-invariant speech recognition model. In particular, they propose fine-tuning a pre-trained model using data from a particular individual.

**Rating:**

Needs Clarification (NC): a submission which does not meet the reviewing criteria and needs clarification for its described problem or solution

**Strengths And Weaknesses:**

The authors present an interesting research idea with compelling motivation from the language learning process. I also appreciated their related works section which did a good job of orienting the reader to the current state of the art in this space. Overall, if they are able to complete some of the modeling work they've set out in the paper I believe this is a strong idea.

The main weaknesses to this paper are that the authors do not present any results and are not specific about the methods they'd like to use. Please see the suggested changes for my thoughts on how to improve in these ways.

**Suggested Changes:**

In future presentations of this work, it would be helpful if the authors were more clear about what kind of model they'd like to fine-tune, and how they would implement the fine-tuning procedure. There is reference to layers and an encoder-decoder, but specifying the model architecture would prove useful for a reader to fully understand their method. (aims: reproducibility, clarity)

In addition, currently the paper does not include any results and thus there are no conclusions to evaluate. From my understanding of the types of papers suitable for the Tiny Papers track, at least some preliminary results are needed.

With this in mind, I'd like to encourage the authors to continue their interesting work.

---

### Official Review · Reviewer_P3og · 2023-03-27

**Confidence:** 3

**Summary Of Contributions:**

The authors propose a technique for speaker-invariant speech recognition tasks. A pretrained model is finetuned on the data from a specific individual, with the goal of reducing the sensitivity to speaker's unique vocal characteristics. To employ the system to multiple individuals, they propose the usage of voice conversion on test-time.

**Rating:**

Needs Clarification (NC): a submission which does not meet the reviewing criteria and needs clarification for its described problem or solution

**Strengths And Weaknesses:**

Strengths:

- Great insights and summarization of ideas (i.e. the grouping of pretext-tasks and the first few lines of the methodology explaining how the representation layers work).

Weaknesses:

- No results were presented. How does this method actually perform on real data? Could you be more specific on how finetuning on a single person's voice would help reduce the its sensitivity to that speaker's unique vocal characteristics?

- I'd like more details both on the related work section about the pretext tasks, as well as in the methodology of your work. Voice-conversion seems to be a big part of the work and it is just briefly spoken about in the last sentences of the methodology.

**Suggested Changes:**

- I'd like to have more details on the related work section, maybe a small sentence to explain the cited works in more detail. Is there something in the literature about how each pretext task impact the learned representations more specifically?

- In the first few sentences of the methodology, you talk about the type of information stored in the layers of the pretrained models. Do you have a reference for those?

- Please add some early results, if available, and more details on the methodology. I'm not sure I understand why finetuning on a single person would enable a speaker-invariant system.

- Also, are there similar work in the literature that uses joint finetuning and voice conversion techniques to learn speaker-invariant models? It'd be nice to add something in this sense as well and maybe compare your results to theirs.

- Please add more details about the voice-conversion needed for inference.

---

### Meta-Review · Area_Chair_2b6C · 2023-04-08

**Recommendation:** Invite to revise
**Confidence:** 4

**Metareview:**

The authors proposed a method for an interesting research idea in speaker-invariant speech recognition. The pros and cons, summarized from the reviewers' comments, are as follows:

Strengths:

1. Great insights and summarization of ideas, including the grouping of pretext-tasks and explanation of representation layers.
2. Interesting research idea with compelling motivation from the language learning process.
3. Well-structured related works section, which orients the reader to the state of the art.

Weaknesses:

1. No results were presented, raising questions about the method's performance on real data.
2. Lack of specificity on how fine-tuning on a single person's voice would reduce sensitivity to that speaker's unique vocal characteristics.
3. Insufficient details on the related work section about pretext tasks and the methodology of the proposed work, particularly voice conversion.

**Summary:**

The authors propose a speaker-invariant speech recognition technique by fine-tuning a pretrained model on individual data and suggest using voice conversion for multiple individuals.

**Comments And Feedback To The Authors:**

I appreciate your contribution towards proposing a technique for speaker-invariant speech recognition tasks. However, it has been pointed out that there is a lack of results, specific details in the methodology and related work sections, and clarity in the paper. I would suggest including more detailed results and elaborating on the methodology, such as the model architecture and fine-tuning procedure. Providing a reference for the type of information stored in the layers of pre-trained models mentioned in the methodology, and clarifying the impact of fine-tuning on a single person's voice, would also be helpful.

**Reason For Not Giving A Higher Recommendation:**

According to the reviewers, the paper lacks clarity and reproducibility, and they have pointed out the absence of results and an ablation study that demonstrate how the proposed methods reduce sensitivity to a speaker's unique voice.

**Reason For Not Giving A Lower Recommendation:**

N/A

---

### Decision · Program_Chairs · 2023-04-08

No revision received; not invited to archive